# Searching for Rheological Conditions for FFF 3D Printing with PVC Based Flexible Compounds

**DOI:** 10.3390/ma13010178

**Published:** 2020-01-01

**Authors:** I. Calafel, R. H. Aguirresarobe, M. I. Peñas, A. Santamaria, M. Tierno, J. I. Conde, B. Pascual

**Affiliations:** 1POLYMAT and Polymer Science and Technology Department, Faculty of Chemistry, UPV/EHU, Avda. Tolosa 72, 20018 San Sebastian, Spain; roberto.hernandez@ehu.eus (R.H.A.); mpenas.3@alumni.unav.es (M.I.P.); antxon.santamaria@ehu.eus (A.S.); 2ERCROS S.A., Innovation and Technology Department, Chlorine Derivatives Division, Diagonal 595, 08014 Barcelona, Spain; mtierno@ercros.es (M.T.); iconde@ercros.es (J.I.C.); bpascual@ercros.es (B.P.)

**Keywords:** rheology, 3D printing, plasticized PVC, buckling, welding

## Abstract

Rheology is proposed as a tool to explore plasticized poly(vinyl chloride) (PVC) formulations to be used in the fused filament fabrication (FFF) 3D printing process and so manufactures flexible and ductile objects by this technique. The viscoelastic origin of success/failure in FFF of these materials is investigated. The analysis of buckling of the filament is based on the ratio between compression modulus and viscosity, but for a correct approach the viscosity should be obtained under the conditions established in the nozzle. As demonstrated by small amplitude oscillatory shear (SAOS) measurements, PVC formulations have a crystallites network that provokes clogging in the nozzle. This network restricts printing conditions, because only vanishes at high temperatures, at which thermal degradation is triggered. It is observed that the analysis of the relaxation modulus G(t) is more performing than the G″/G′ ratio to get conclusions on the quality of layers welding. Models printed according to the established conditions show an excellent appearance and flexibility, marking a milestone in the route to obtain flexible objects by FFF.

## 1. Introduction

Additive manufacturing (AM), also generally known as 3D printing, allows the fabrication of fully customized objects with a high geometric complexity and with a significant reduction of both time and manufacturing costs. In addition, this technology follows the “Do it yourself” (DIY) trend, which implies that not only professional manufacturers but also customers can produce and/or modify their own designs. Various AM processes have considerably improved in last three decades, becoming established in the commercial market [1,2,3,4,5,6,7,8,9]. Each process depends on different phenomena, specific materials, and their physical and rheological characteristics [10,11,12,13,14,15]. Among the materials used for AM polymers have become a main center of interest for a wide range of applications, in addition to metals [16] and ceramics [17]. Versatility and synthetic adaptability have made polymers the most used for AM processes [18,19,20,21,22,23,24,25,26,27,28,29,30,31,32,33]. Fused deposition modeling (FDM) or so-called fused filament fabrication (FFF) is one of the most popular AM processes due to its simplicity. In this process, the object is generated by depositing layer-by-layer the melted material, which flows through a nozzle, in a predefined tool path [11,34]. Although currently new feedstock devices are being developed, the procedure based on filament feedstock remains the favorite. So far, the most popular polymers utilized for FFF are polylactic acid (PLA) and acrylonitrile-butadiene-styrene copolymer (ABS). Polycarbonate (PC), poly (ether imide) (PEI), Nylon 12, thermoplastic polyurethane (TPU), and other primarily amorphous thermoplastic polymers are also used to achieve extremely complex geometries. Except TPU, which allows 3D printing of ductile parts, currently the polymers used are brittle, which limits their applications for certain purposes. The exponential technological progress and the growing demands of the market yield the higher motivation to develop new materials for FFF. Among these new materials, the inclusion of more polymers flexible and ductile (in addition to TPU) is being considered day by day.

Notwithstanding, in FFF there are several obstacles in the development of new materials, such as the size and the mechanical properties required for the filament. A proper diameter can be achieved refining the filament fabrication methods. However, furthermore, a filament to be used in the FFF process should be sufficiently rigid to withstand the force applied by the counter-rotating rollers to act as a piston in the extrusion process without buckling failure. The currently used materials have a glass transition temperature (*T_g_*) above room temperature or sufficient degree of crystallinity to impart substantial strength to the filament to avoid buckling. Flexible materials, however, are commonly amorphous with a *T_g_* close to room temperature. This characteristic causes that the filament does not ensure sufficient strength to drive the melt through the nozzle and, so, filament buckling is produced. This problem is even greater for some specific configurations, as in Bowden type printers, where the gears are located far from the printing head. According to the model proposed by Venkataraman et al. [35], the filament will buckle if the pressure applied by the rollers exceeds that of the material’s critical buckling stress. The authors estimated that this critical parameter depends on some specific geometrical and flow settings of the process. Nevertheless, this model predicts the filament buckling (failure) but does not account for printability of the system, which is actually closely linked to the flow and viscoelastic properties of the material. The rheology meets the industry’s need for knowledge to develop new materials for FFF, because it responds to the processes that take place during the printing stages, such as, viscous flow in the nozzle and viscoelastic behavior during interlayer welding [36,37,38]. In this sense, the rheology is again conformed as a rapid and effective screening tool to predict the success or failure of novel materials in the whole 3D printing process, both extrusion and welding, thus avoiding the current laborious, time consuming and non scientific trial-and-error methodologies.

In this paper, rheology is proposed as a tool to explore plasticized PVC formulations to be used in the FDM process, opening the possibility of manufacturing flexible and ductile objects by this technique. This is not an elementary task, because PVC formulations face two important deficiencies: (a) poor thermal stability due to thermo-mechanical degradation at typical processing temperatures [39] and (b) particular flow properties assigned to the formation of crystallites (even in plasticized samples) that give rise to a physical network, which only disappear at high temperatures [40]. These shortages have limited the introduction of PVC compounds in the FFF market. In a recent paper [41] we have demonstrated the feasibility of PVC/PBA (polybutylene acrylate) copolymers, in which the flexible PBA acts as a plasticizer, to be used in 3D printing of pellets, but not by the FFF method. Therefore, our present work constitutes a new step for the introduction of PVC flexible formulations, and so ductile materials, in additive manufacturing and fused filament fabrication in particular. The paper is organized as follows: (a) preparation and characterization of PVC/DINP (diisononyl phthalate) formulations, (b) analysis of buckling conditions and flow/clogging in the nozzle, based on the rheological results, (c) analysis of interlayer welding according to viscoelastic data, and (d) elaboration of flexible objects by FFF with the selected compositions and printing conditions.

## 2. Materials and Methods

### 2.1. Samples Preparation

Plasticized PVC/DINP samples were supplied by Ercros S.A. (Tarragona, Spain) and were based on a commercial grade Etinox^®^ 650 resin. The blends were obtained by melt mixing of the PVC resin with 2 phr of a Ca/Zn stabilizer and 1 phr of epoxydated soy bean oil wherein ‘‘phr” is an abbreviation for ‘‘parts per hundred parts of resin” and the appropriate amounts of diisononyl phthalate (DINP). The fusion and homogenization of the formulations was carried out in a two-roll mill at temperatures between 120 and 140 °C and pressed at the same temperature to mold the samples for the corresponding tests. Subsequently sheets were cut into pellets. The characteristics of the plasticized PVC/DINP compounds used in this work are summarized in Table 1.

### 2.2. Characterization

An Instron universal testing machine (Instron 5569, Norwood, MA, USA) was used to measure the compression modulus of the PVC/DINP compositions. At least 5 prismatic specimens of 24 × 6 × 6 mm^3^ were used for each compression stress–strain test. Tests were carried out at room temperature and at displacement rate of 20 mm/min. The compression modulus was defined as the initial linear modulus.

A Dynamic Mechanical Thermal Analyzer, Triton 2000 DMA from Triton Technology, was used in bending deformation mode to carry out a mechanical dynamic thermal analysis (DMTA). The samples were heated from −100 to 120 °C at a constant heating rate of 4 °C/min and a frequency of 1 Hz. The tests were performed at low strain amplitudes to obtain the elastic modulus (E’) and the loss tangent (tan δ) ensuring a linear viscoelastic response. These measurements allowed detecting the glass transition temperature, *T*_g_, given by a maximum peak in loss tangent, tan δ.

The viscosity of the melts was evaluated by capillary rheometry. Extrusion flow experiments were performed in a Göttfert Rheo-Tester 1000 rheometer using a capillary die of L/D = 30/1 and a zero length capillary, to obtain the viscosity curves at the temperatures and shear rates indicated in the Results and Discussion section.

Small amplitude oscillatory shear (SAOS) measurements in the molten state were carried out using a AR-G2 TA rheometer with a parallel-plate fixture (25 mm diameter), conducting experiments in the linear regime. The shear elastic modulus, G′ and the shear loss modulus, G″ were evaluated as a function of temperature at successive constant frequencies of 0.1; 0.21; 0.46; 1; 2.15; 4.64 and 10 Hz.

Stress relaxation experiments were carried out in a strain controlled ARES TA viscoelastometer at the temperatures indicated in the Results and Discussion section. The shear relaxation modulus, G(t) = σ (t)/γ, were σ (t) is the time dependent stress and γ the constant strain, was obtained in the linear viscoelastic regime.

Scanning electron microscopy (SEM) was used to analyze the surface morphology and cohesive aspect of the printed objects. TM3030Plus Tabletop Hitachi electron microscope (Hitachi High-Technologies Corporation, Japan) was employed that operated at 15 kV in a standard (SD) observation mode. Prior to observation, the printed samples were cryo-fractured in liquid nitrogen and covered with a fine gold layer.

### 2.3. Filament Preparation

Filaments to be used for FFF 3D printing were prepared in a Haake MiniLab twin-screw extruder (Thermo Fisher Scientific, Australia) at a temperature of 130–150 °C, 100 rpm screw speed, using a diameter die of 1.75 mm. Obtained filaments presented a 1.70 mm ± 0.05 mm diameter.

### 2.4. Filament Fusion Fabrication Printing

The printing process was carried out in a Bowden type Voladora NX printer (Tumaker, Spain) with a 0.4 mm nozzle. The printing geometries were originally designed in Solidworks 2016 Editor and printed using Simplify printing program. Printing conditions including printing temperature, printing substrate temperature, printing velocity and infill density are detailed in the Result and Discussion section.

## 3. Results and Discussion

### 3.1. Materials Characterization

Different amounts of DINP were added to PVC in order to tailor the mechanical and rheological performance of selected samples. Thus, as can be seen in Table 1, the hardness of samples decreases as the plasticizer amount increases. From an industrial point of view, a polymer could be considered flexible when the Shore A hardness value is less than 90. As can be seen in Table 1, the formulations with 30 and 40 wt% of DINP content present values of Shore A below 90; however, the formulations with 20 and 10 wt% present values of Shore A close to 90. Therefore, we face with two flexible formulations (PVC/40DINP and PVC/30DINP), one semi-rigid (PVC/20DINP) and a rigid one (PVC/10DINP).

The thermomechanical and rheological behavior of the samples was also analyzed using DMTA and capillary rheometry. In Figure 1a DMTA results of different PVC/DINP samples are presented, showing the dependence of the loss tangent, tan δ, with DINP content (neat PVC is also displayed for comparison purposes). The loss tangent peaks are well defined, marking a relaxation, which is associated with the glass to rubber transition temperature, *T_g_*: As is expected, the addition of DINP shifts the *T_g_* of PVC to lower temperatures (Table 1). The flexibility determined in terms of Shore A can directly be correlated to the values of the glass transition temperature. PVC/10DINP shows the higher *T*_g_ value and it is observed the sample is brittle at room temperature.

In Figure 1b the results of the apparent viscosity vs. apparent shear rate at a temperature of *T* = 180 °C are presented. Therefore, these data have not been corrected from pressure losses at the ends of the capillary, neither from non-Newtonian behavior. This issue is discussed in the next section. As shown, the samples present a remarkable shear thinning behavior. As expected, the incorporation of higher amounts of plasticizer results in viscosity decrease. The data are fitted to the Cross-Yasuda model [42] taking *η_∞_* = 0:(1)$$\eta \left(\dot{\gamma }\right)=\frac{\eta_{0}}{{\left(1+\lambda \dot{\gamma }\right)}^{\alpha }},$$
where *η*_0_ is the Newtonian viscosity, *λ* is a relaxation time linked to the onset of non-Newtonian behavior and *α* a shear thinning index.

### 3.2. Rheological Analysis

The FFF process demands specific requirements to the new materials in order to avoid inherent problems of this technology, such as inaccurate filament diameter, backflow, and, in the case of flexible materials, buckling in particular [11,13]. Thus, a suitable feedstock material should provide an accurate balance between the physical properties in the solid state, mainly the compression modulus (K), and the rheological properties in the molten state. The proposed model follows the analytical procedure reported by Venkataraman et al. [35] to describe the buckling effect. The FDM process configuration can be described as depicted in Figure 2:

Where *R* is the filament radius, *L* is the length between the rollers and the printing head, and *l* and *r* are the radius and the length of the nozzle. On the one hand, in the solid state, the buckling criterion for elastic column is given by Euler’s analysis for two pin ended boundary conditions as [43]:(2)σcr=Kπ2(LR)2,
where *σ_cr_* is the critical buckling stress, *K* is the compressive modulus for FFF filaments, *L* is the length of the column (in this case, the length of the filament between the rollers and the top of the hot section of the nozzle), and *R* is the radius of the filament. The quantity *L/R* is called the *slenderness ratio* of the filament. Equation (2) shows that the critical stress is directly proportional to the compression modulus of the material, and inversely proportional to the square of the slenderness ratio of the filament. On the other hand, in the molten state, the pressure drop (∆*P*) to drive a non-Newtonian fluid through a capillary of length *l* and radius *r* is given by:(3)$$\Delta P=\frac{8Q\eta_{a}l}{\pi r^{4}},$$
where *η_a_* is the apparent viscosity determined using capillary rheometry and *Q* is the volumetric flow rate. The volumetric flow rate can be calculated in terms of printing velocity, assuming the mass conservation law, as *Q* = *πR*^2^*V_p_*. Thus, in order to avoid bucking in the 3D printer, the pressure required to produce the flow should be smaller than the critical stress for buckling, ∆*P* < *σ_cr_*. Therefore, from Equations (2) and (3) the following equation can be deduced:(4)$$\frac{K}{\eta_{a}}>\frac{8Ql{\left(L/R\right)}^{2}}{\pi^{3}r^{4}}.$$

According to this equation, for a given printer configuration geometry and flow rate, filaments with a value of *K*/*η_ap_* higher than a certain critical value will be suitable for FFF avoiding buckling. It is to note that the higher the *K*/*η_ap_* value for a specific feedstock material, the better the printability. This equation has been developed for Newtonian fluids. However, in the case of shear thinning materials, additional considerations should be taken into account.

On the one hand, it is necessary to bear in mind that capillary rheometry produces a pressure-driven flow that deviates from a parabolic velocity profile when the fluid is no Newtonian. The shape of the profile is defined by the power-law index *n* < 1; the smaller *n* is, the more the profile deviates from the parabolic shape. Thus, shear rate at the wall, ${\dot{\gamma }}_{w}$, can be corrected using the Weissenberg–Rabinowitsch–Mooney equation [44]. For a circular capillary die, the shear rate at the wall is:(5)$${\dot{\gamma }}_{w}=\frac{4Q}{\pi r^{3}}\left(\frac{3n+1}{4n}\right),$$
where *Q* is the volume flow rate and *r* is the capillary radius.

On the other hand, it is also noteworthy that, in capillary rheometry, viscoelastic fluids tend to form re-circulating corner vortices at the entrance of the capillary, where the fluid stretches and accelerates thought a sudden contraction. The more elastic is the melt, the greater is the influence of the corner vortices. The apparent shear stress at the wall ($\sigma_{wa}$) is calculated from the pressure (or actually from the difference between the atmospheric pressure and barrel pressure, ∆*p*) measured in the barrel:(6)$$\sigma_{w}=\frac{∆p-∆p_{e}}{2\left(l/r\right)},$$
where *l* is the capillary length an *r* is the capillary radius and ∆*p_e_* is the extra pressure drop. These effects are considered negligible for capillaries with *l/d* ratio above 30. However, in 3D-printing, the nozzle is usually very short (*l*/*r* < 2), so the extra pressure drop arising at the entrance of the capillary die should be taken into account. Correction is commonly carried out through the so-called Bagley plots [44], where several capillaries with the same diameter and different lengths are used. However, in this case, ∆*p_e_* has been measured directly using a “zero-length capillary”.

Therefore, on the sake of the real conditions in 3D printing, the apparent viscosity, *η_ap_*, considered in the *K*/*η_ap_* ratio, should be corrected through Weissenberg–Rabinowitsch–Mooney correction and entrance pressure correction, in order to consider the shear thinning behavior of the polymeric melt and the viscoelastic effects at the entrance of the capillary.

In Figure 3 the ratio between the compression modulus to the viscosity (*K*/*η*), with *K* measured at room temperature (Table 1) and viscosity at 180 °C, of different samples is presented as a function of shear rate, both corrected and uncorrected. An analysis of the corresponding velocity of the polymer melt in the nozzle, as compared to the printing velocity fixed in the corresponding printing conditions, shows a matching between both. This allows linking directly the printing velocity to the shear rate in the nozzle. Dotted lines show the critical values calculated between the maximum and minimum print velocity (100 and 0.1 mm/s), for a length between rollers and the top of the heated nozzle of *L* = 60 mm, a nozzle length of *l* = 0.5 mm and two different nozzle diameters: 0.4 and 1.2 mm. The corresponding shear rate limits are 2000 s^−1^ and 2 s^−1^ for the 0.4 mm nozzle and 666 s^−1^ and 0.66 s^1^ for 1.2 mm nozzle.

Based on this model, the PVC/10DINP and PVC/20DINP formulations could be printed as they presented values greater than the critical one, regardless of the diameter of needle used. However, as it was discussed below, the PVC/10DINP formulation led to a brittle filament that breaks when trying to load in the printer, resulting in a print failure. On the other hand, the PVC/30DINP and PVC/40DINP filaments present lower *K*/*η_ap_* values than the critical ones if a nozzle of 0.4 mm diameter was used, indicating that the material should buckle. However, with a 1.2 mm diameter nozzle, the PVC/30DINP could be fairly processed at printing velocities corresponding to shear rates close to 10 s^−1^.

As can be seen in Figure 3b, the corrected viscosity resulted in an increase in the ratio *K*/*η*, which led to an enhanced processing range. This increase in the ratio *K*/*η* is due to the aforementioned both corrections that give rise to a true viscosity value lower than the apparent one. The available processing shear rate increased up to 100 s^−1^ for the PVC/40DINP filament when it was used a 1.2 mm diameter nozzle, and up to 30 s^−1^ for the PVC/30DINP. However, although the correction enhanced the processing range, the changes were not as significant as required to justify the behavior during the extrusion process explained below.

These theoretical printing characteristics were compared with experimental printing results. As can be seen in Table 2 and Figure 4 materials were nicely printed at temperatures above 200 °C, but the printability of the samples was limited below this temperature. In fact, only the materials presenting high amounts of plasticizer can be printed at 180 °C. This was not initially expected, because the samples indeed flow in the capillary rheometer at this temperature, as can be seen in Figure 1. However, we have to consider that the diameter of the nozzle is actually the half of that of the capillary rheometer. More relevant can be the different thermal history of the melt in both devices, the rheometer and the printing nozzle: in the former the measurements were made under settled isothermal conditions, whereas in the latter the data were probably taken under transient thermal conditions.

On the other hand, the results of Figure 3 highlight a fundamental mismatch among the theoretical predictions and the experimental printing conditions. Therefore, it is necessary to take into account several approaches not considered so far.

First, when trying to print at 180 °C, the printing failed not because of material bucking but because of nozzle clogging. Therefore, the processing window was not just determined by filament buckling: additional viscoelastic characteristics should be considered. Thus, temperature sweep experiments, at different constant frequencies, were performed by SAOS. Figure 5 shows the G′ and G″ results as a function of temperature for the sample PVC/40DINP at respective frequencies of 0.1, 1 and 10 Hz. The rest of the frequencies have been omitted to avoid data mess. Besides of the observed maxima for both moduli, which was commented below, we remarked the predominantly solid behavior, with G′ > G″, up to approximately 180 °C. Actually, a temperature that marks a transition from solid (G′ > G″) to liquid (G″ > G′) can be defined as the temperature at which G′ = G″.

Certainly, such a transition cannot be properly defined considering only one frequency, as was highlighted by Winter [45]. As frequency increased a shorter time was involved in the test, which should imply a higher temperature for the transition. Interestingly enough, our results of Figure 6, which shows the *T*_(*G′* = *G″*)_ transition of each sample at different frequencies, indicate that the effect of frequency was not indeed significant. The analysis of this unexpected result was out of the scope of this work.

In the aim to focus rather on the effect of samples composition on the observed solid–liquid transition, we considered the results taken at a frequency of 1 Hz, as can be seen in Figure 7. The corresponding *T*_(*G′* = *G″*)_ transition values were included in the Table 3. Considering previous studies on the particulate flow of PVC formulations [40] we assumed that the results were a consequence of the physical network formed by crystallites that only vanish at high temperatures, that is to say at temperatures at which G″ > G′ was observed. The crystallites network impeded reaching the terminal/flow viscoelastic zone. However, in the interval 180–210 °C (Figure 7), depending on composition, a cross-over of dynamic moduli occurred and the flow zone, characterized by G″ > G′, was attained. On the other hand, a G′ and G″ maxima was observed upon increasing the temperature in the range 120–170 °C. As far as we know this result has not been reported in literature and can be explained considering the interaction between the plasticizer and PVC, which leads to structural changes during the fusion of the samples [46].

We suggest that the predominant elastic behavior (G′ > G″) of the formulations at low temperatures, owed to crystallites network, hinders the flow through the nozzle and produces clogging. From the results shown in the figure and the corresponding table, it was deduced that the cross-over temperature of PVC/DINP mixtures increased as the DINP amount decreased. Interestingly, the corresponding cross-over temperatures matched the lowest printing temperature for these systems. Therefore, we found that in addition to fulfill the buckling criterion, the material should present a predominantly viscous behavior when printing, in order to avoid nozzle clogging.

In addition to these rheological issues, it is also necessary to analyze the problems that can derive from the thermal instability of the samples. In the case of PVC derivatives, material degradation is a point that should be necessarily considered. Thus, if the extrusion rate is too low, the material would be exposed at high temperatures for an excessive time, promoting the thermal degradation of the sample. This restricts both, processing temperature and printing velocity. This issue also limited us to obtain viscosity results at the corresponding printing temperatures, above 180 °C. In any case, it can be assumed that an increase in temperature will reduce the materials viscosity helping to avoid buckling and clogging.

### 3.3. Welding between Layers

Another key point of additive manufacturing processes is the interlayer adhesion or bonding between layers, since it determines the quality, in terms of mechanical properties, of the printed specimens. This process occurs through molecular inter-diffusion of polymer chains across the interface between two layers in contact. Thus, this property is strongly dependent on the viscoelastic features of the materials at the welding temperature, which in turn depend on printing velocity and temperature of the printing support [37,47]. Considering that a suitable inter-diffusion is necessary for a successful bonding, the response of the material at the welding temperature should correspond to the terminal or flow regime, i.e., in the region where G″ > G′. That is to say, interlayer adhesion is favored by a predominantly viscous behavior of the material at the welding temperature, which is always lower than the printing temperature. Even so, materials with a predominantly solid linear viscoelastic behavior (G′ > G″) have shown adhesion properties in different applications [48]. Face to this apparent contradiction, the Dahlquist criterion [49], developed for adhesives has been considered for 3D printing [50]. This rule establishes a critical elastic modulus value of 3 × 10^5^ Pa (corresponding to a compliance value, J_c_, of 10^−5^ Pa^−1^) below which a material shows good tack or instantaneous adhesion, regardless the prominence of G″ over G′ state. [51,52]. The adhesion temperature is, of course, related to the cooling time of the samples, which depends not only on the printing conditions, but also on the thermal conductivity of each material. Measuring the actual time and the temperature at which the welding occurs in our materials is out of the scope of this work. The work done by Seppala et al. [37,47], estimating a cooling time less than 1 s for ABS copolymer can be a good reference, considering that thermal conductivity is very similar for all the polymers.

To take advantage of the Dahlquist criterion, stress relaxation experiments were performed in shear in order to determine the minimum time needed for welding in our samples. As a reference, a welding time of 1 s (that of ABS) was also plotted.

Results depicted in Figure 8a indicate that the higher the DINP content, the easier the Dahlquist criterion is achieved. Thus, PVC/40DINP can bring about good adhesion even at welding temperatures of 100 °C. In contrast, PVC/20DINP samples barely reach the welding zone even at 180 °C, announcing poor welding characteristics. These results showed that, in order to ensure a correct interlayer adhesion, the material should be at sufficient high temperature at least for 1 s. This fact limits the lower printing temperature, as can be seen in the bending tests shown in Figure 8b). Sample bars printed near the lowest printing temperature show poor welding and, so, printing failure. This is compatible with the viscoelastic behavior of the printing filament, which is not appropriate to ensure the material welding in terms of the Dahlquist criterion. However, increasing the printing temperature gave rise to a good welding, which is noticed at the microscopic scale (Figure 8c). Notwithstanding in this work we have considered the Dahlquist criterion as a representative parameter for a successful interlayer welding, a deeper work in this area should contemplate further parameters related to chain entanglement/disentanglement process during printing.

## 4. Conclusions

A thermal, rheological and viscoelastic analysis of PVC formulations plasticized with DINP was carried out, with the aim of obtaining useful parameters for a sound evaluation of the feasibility of these formulations for the FFF procedure. Considering that these formulations were flexible (Shore A hardness values below 90), this work lay in the more general context of obtaining flexible/ductile objects by 3D printing.

The issue of filaments buckling, which is typical of flexible polymers, is treated considering the limits marked by the ratio of the compressibility of the filament in the solid state (room temperature) to the viscosity in the molten state (during flow in the nozzle). The viscosity was determined considering the actual printing conditions, which implies shear rates in the region of shear thinning, and very short capillaries, which gave rise to significant entrance pressure effects. Only taking into account these corrections accurate plots for buckling/no buckling maps were achieved. Nevertheless, we remarked that even in the absence of buckling printing failed at *T* = 180 °C because of nozzle clogging. By means of SAOS measurements we demonstrated that the origin of clogging lay on the crystallites network that was revealed by a predominant elastic behavior, G′ > G″. At high temperatures, which depend of DINP concentration, the network vanished as G″ > G′ was observed, giving rise to a suitable flow in the nozzle and good printing conditions.

Printed models show an excellent appearance and flexibility, marking a milestone in the general attempts to obtain flexible objects by FFF. An analysis of the viscoelastic results allowed us to conclude that the Dalquist’s criterion is probably more appropriate than G″ > G′ consideration to investigate the quality of layers welding.

## Figures and Tables

**Figure 1 materials-13-00178-f001:**
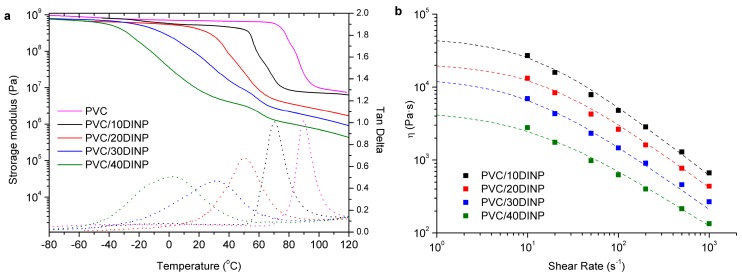
(**a**) Elastic modulus and loss tangent as a function of temperature at a constant frequency of 1 Hz. (**b**) Viscosity as a function of shear rate at 180 °C. The compositions are indicated.

**Figure 2 materials-13-00178-f002:**
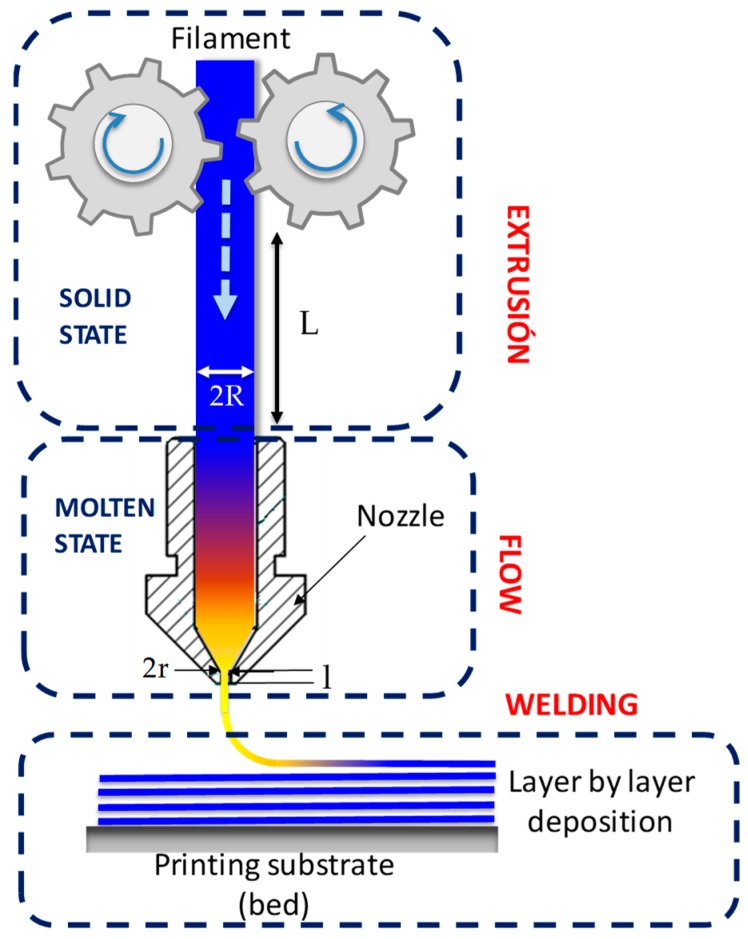
Geometric considerations for the occurrence of buckling.

**Figure 3 materials-13-00178-f003:**
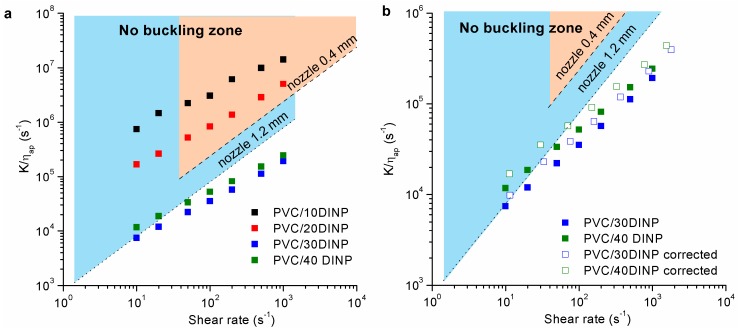
(**a**) Ratio *K*/*η_ap_* vs. shear rate with the viscosity taken at 180 °C. The dotted lines correspond to the critical value using nozzles of 0.4 mm (…..) and of 1.2 mm (- · - · - · -). (**b**) Detail of the corrected and uncorrected *K*/*η_ap_* values by applying the Weissenberg–Rabinowitsch–Mooney and Bagley corrections to the melt viscosity for PVC/30DINP and PVC/40DINP filaments.

**Figure 4 materials-13-00178-f004:**
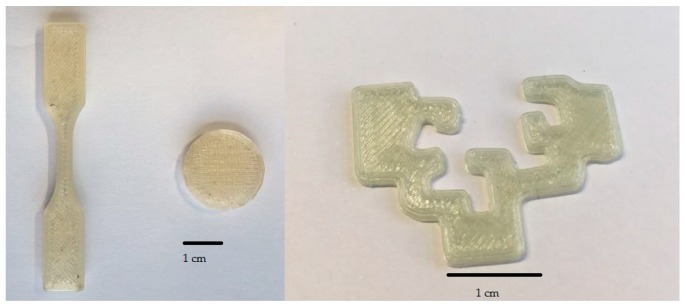
Printed models for PVC/40DINP sample.

**Figure 5 materials-13-00178-f005:**
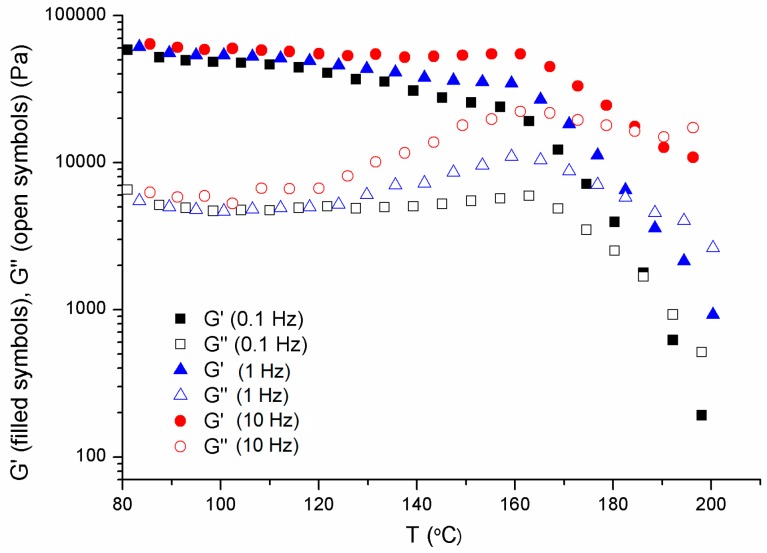
Shear elastic (G′) and viscous (G″) moduli vs. temperature taken at constant corresponding frequencies of 0.1, 1, and 10 Hz for the sample PVC/40DINP.

**Figure 6 materials-13-00178-f006:**
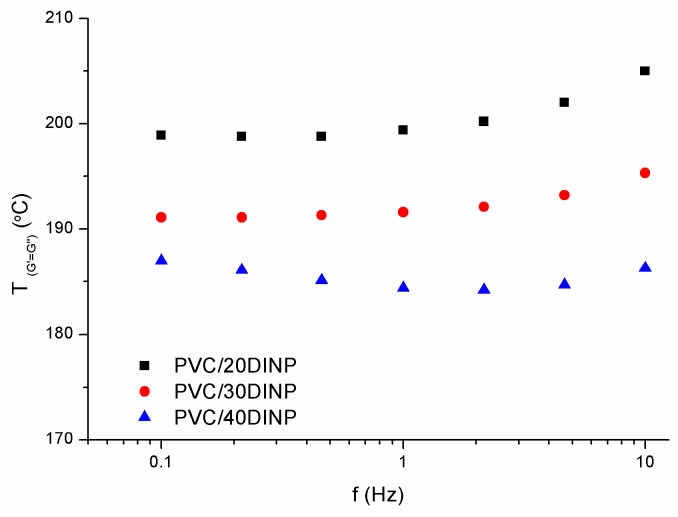
Values of the cross-over G′ = G″ temperature, *T*_(*G′* = *G″*)_, as a function of the applied frequencies for the different samples.

**Figure 7 materials-13-00178-f007:**
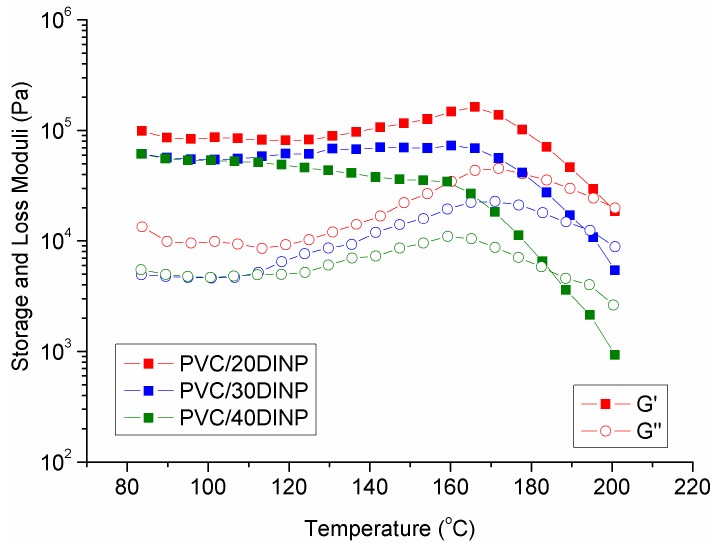
Shear elastic (G′) and viscous (G″) moduli vs. temperature taken at a frequency of 1 Hz for printable formulations.

**Figure 8 materials-13-00178-f008:**
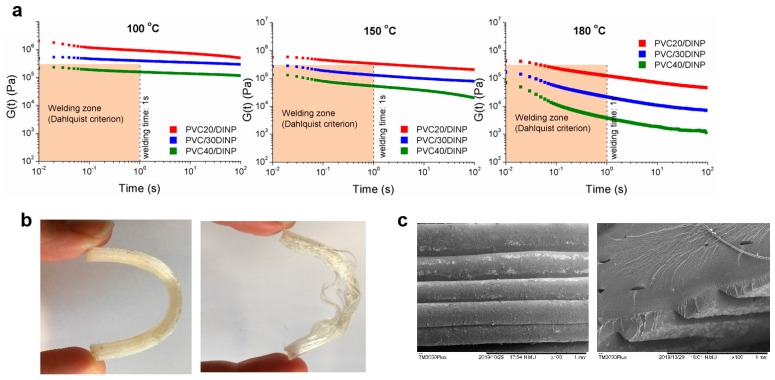
(**a**) Stress relaxation experiments at different temperatures for printable samples, showing welding zone according to Dahlquist criterion (see text). (**b**) Bending test to show good (left) and poor (right) layers adhesion in samples obtained at two different printing temperatures for PVC/40DINP: 210 °C (left) and 190 °C (right). (**c**) SEM micrographs for PVC/40DINP printed at 210 °C.

**Table 1 materials-13-00178-t001:** PVC/DINP samples prepared by mixing.

Sample	DINP (%)	Shore A/D	*T*_g_ (°C) ^1^	*K* (MPa) ^2^
PVC/40DINP	40	69/20	1	32 ± 2
PVC/30DINP	30	87/38	30	51 ± 1
PVC/20DINP	20	95/65	50	784 ± 99
PVC/10DINP	10	96/82	60	2190 ± 250

^1^*T*_g_ determined by DMTA as explained in the text. ^2^ Compression modulus (*K*) determined as explained in the Experimental Section.

**Table 2 materials-13-00178-t002:** Printing temperatures for different PVC/DINP mixtures.

	Print Temperature (°C)			
Sample	180	190	200	210
PVC/40DINP	-	✓	✓	✓
PVC/30DINP	-	-	✓	✓
PVC/20DINP	-	-	✓	✓
PVC/10DINP	brittle			

**Table 3 materials-13-00178-t003:** Cross-over (G′ = G″) temperatures, *T*
_(*G′* = *G″*)_, for printable formulations obtained at frequency of 1 Hz.

Sample	*T* _(*G′* = *G″*)_
(°C)
PVC/10DINP	-
PVC/20DINP	199
PVC/30DINP	192
PVC/40DINP	185

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
