# Peer review of "Searching for Rheological Conditions for FFF 3D Printing with PVC Based Flexible Compounds"

_materials, 2020, doi:10.3390/ma13010178_

Round 1

Reviewer 1 Report

The manuscript is an experimental study comparing the rheological response of PVC formulations with their performance in a Fused Filament Fabrication (FFF) 3-D printer. PVC formulations containing different amounts of diisononyl phthalate (DINP), corresponding to different rigidity in the solid state and different rheology response in the melt state, are analyzed by Dynamic Mechanical Analysis (DMA), capillary viscosimetry, Small Amplitude Oscillatory Shear (SAOS) and linear stress relaxation rheometry and Scanning Electron Microscopy (SEM).

The authors try to find quantitative relationships between the independent mechanical measurements and the FFF printability. In spite of the efforts made, however, I think that the such an attempt is not successful. There are many flaws in the discussion that should be reconsidered. I try to list them:

1) The capillary viscometry data are not clearly presented. The authors say that they are able to make both the Bagley (as they use both a zero length and a L/D=30 capillary) and the Mooney correction to their pressure measurements. They do not specify, however, if Figure 1 refers to corrected or uncorrected data;

2) When applying the Venkataraman procedure to assess printability based on the ratio between compression modulus and viscosity the authors produce the plots reported in Figure 3, where the above ration is plotted as a function of shear rate. But they say nothing about the shear rate reached into the nozzle. The latter should be easily estimated from the polymer flow rate and the nozzle geometry. In the paper there is no trace of this information, which could be useful to better understand the claimed discrepancy between the mechanical measurements and the actual printability of the samples. Incidentally, there is a mistake in Eq.(3), where the radius should be elevated to the fourth power, not to the second power. Probably the error is not influent because in the subsequent Eq.(4) the correct power is reported;

3)  On page 9 (lines 286-301) the authors use the G'/G" ratio at 1 Hz to justify that (quote) "...PVC/DINP mixtures showed a predominantly solid behavior even at very high temperatures, near 180°C, with an elastic modulus over the viscous modulus, G’>G’’. This is a consequence of the remaining physical network formed by crystallites in PVC formulations that impedes reaching the terminal or flow viscoelastic zone". This claim cannot be accepted, for several reasons.

First, the authors' statement is contradicted by their same experimental measurements. The capillary rheometry measurements reported in section 3.1 clearly show that the all polymer samples indeed flow at 180°C. Notice that the capillary geometry is very similar to that of the nozzle. The immediate question, therefore, is: Why is flow regular in the capillary and not regular in the nozzle, given the same temperature and a similar geometry?

Second, the authors use the G"/G' ratio argument in a very superficial and wrong way: polymer melts, that is, polymers in a fully liquid state, do exhibit values of G'>G". As any rheologist knows, it is only a matter of frequency. At high frequencies (above the so-called cross-over frequency) one has G'>G", whereas G'<G" is obtained below the cross-over. This means that, if the authors had used a lower frequency (say, 0.1 Hz) instead of 1Hz, they could have found G'<G", thus contradicting their own argument. The point here is that the solid-like or liquid-like beahviour of a polymer melt cannot simply be determined by measurements at a single frequency! As a consequence, the information contained in Figure 5, and the following discussion, are meaningless.

Third, Figure 5 is indicative of an anomalous behaviour, as both moduli go through a well defined  maximum upon increasing the temperature. Did the authors investigate the reason for the observed behaviour?

As a more general comment to this point, my personal opinion is that the rheological and printing information cannot be directly compared because the THERMAL HISTORY is relevant, and not the temperature. In capillary rheometry, for example, the data are typically taken under constant, uniform temperature. In the SAOS experiments the thermal history is not reported at all. Finally, in the printing process the situation is also not defined, and in any event is expected to be completely different from both capillary and SAOS rheology, as the flow through the nozzle obviously takes place under quite fast transient thermal conditions. Maybe (but this is just a hypothesis) the characteristic flow time is much shorter than the time necessary for a full melting of the polymer at 180°C. In any case, I am not surprised to see the discrepancies between rheology and processing. Certainly, based on the above, the G'>G" criterion cannot be accepted;

4) The stress relaxation experiments are confusing and somewhat naive. They are performed at three different temperatures (100°C, 150°C and 180°C), then (see the caption of Figure 6 and the text at lines 335-347) the Dahlquist criterion is applied by assuming a welding time of 1 second. Although the procedure is qualitatively reasonable, it seems very difficult to me that it could be translated into a quantitative criterion, as the authors do not measure the filament temperature history from nozzle exit to welding. I understand that this can be a formidable task. For the same reason, however, trying to mix quantitative results with qualitative indications is, as I said at the beginning of this point, naive at best.

In summary, my opinion is that the manuscript cannot be accepted in its present form. There are unclear points, and the use of some rheological concepts is literally wrong.

Author Response

Dear Editor

We are pleased in submitting to your consideration for publication in MATERIALS a revised version of our manuscript entitled Searching for rheological conditions for FFF 3D printing with PVC based flexible compounds.

All the concerns and requirements of the reviewers have been contemplated and the manuscript has been modified accordingly (changes are highlighted in red in the text). Detailed corrections are listed as follows (the sentences given in Italic corresponds to the changes introduced in the revised manuscript):

Reviewer 1

The reviewer states that the capillary data are not clearly presented and asks if Figure 1 refers to corrected or uncorrected data.Answer: This has been clarified in the revised version including the following paragraph:In Figure 1b the results of the apparent viscosity vs. apparent shear rate at a temperature of T= 180ºC are presented. Therefore, these data have not been corrected from pressure losses at the ends of the capillary, neither from non-Newtonian behavior. This issue is discussed in the next section.       The reviewer observes that the shear rate reached in the nozzle is not estimated considering the flow rate and the nozzle geometry and notes a mistake in Eq. 3.

Answer: The mistake in equation 3 has been corrected. On the other hand, in the new version we explain that the actual shear rate in the nozzle is calculated assuming that the printing velocity coincides with the velocity of the melt in the noodle, as has been proved. The following paragraph has been included:

An analysis of the corresponding velocity of the polymer melt in the nozzle, as compared to the printing velocity fixed in the corresponding printing conditions, shows a matching between both. This allows linking directly the printing velocity to the shear rate in the nozzle

Our hypothesis of G’>G’’ being a consequence of the remaining physical network formed by crystallites is not accepted by the reviewer for several reasons detailed below. The reviewer finds a contradiction in the fact that 3D printing cannot be carried out at T=180ºC, although the samples flow in the capillary rheometer at this temperature.

Answer: We think that this is rather an apparent contradiction, because the flow conditions are not the same in both devices. A new paragraph has been included to explain this in the revised version:

As can be seen in Figure 4 materials were nicely printed at temperatures above 200ºC, but the printability of the samples was limited below this temperature. In fact, only the materials presenting high amounts of plasticizer can be printed at 180ºC. This was not initially expected, because the samples indeed flow in the capillary rheometer at this temperature, as can be seen in Figure 1. But we have to consider that the diameter of the nozzle is actually the half of that of the capillary rheometer. More relevant can be the different thermal history of the melt in both devices, the rheometer and the printing nozzle: in the former the measurements are made under settled isothermal conditions, whereas in the latter the data are probably taken under transient thermal conditions

The reviewer claims that the solid-like to liquid-like transition temperature cannot simply determined by measurements at a single frequency.

Answer: Obviously the reviewer is right. Considering the scope of the paper and for the sake of simplicity we did not included the analysis of the effect of the frequency on the transition temperature. This has been corrected in the revised version, including the following text and figures:

First, when trying to print at 180ºC, the printing fails not because of material bucking but because of nozzle clogging. Therefore, the processing window is not just determined by filament buckling: additional viscoelastic characteristics should be considered. Thus, temperature sweep experiments, at different constant frequencies, were performed by SAOS. Figure 5 shows the G’ and G’’ results as a function of temperature for the sample PVC/40DINP at respective frequencies of 0.1, 1 and 10 Hz. The rest of the frequencies have been omitted to avoid data mess. Besides of the observed maxima for both moduli, which is commented below, we remark the predominantly solid behavior, with G’>G’’, up to approximately 180ºC. Actually, a temperature which marks a transition from solid (G’>G’’) to liquid (G’’>G’) can be defined as the temperature at which G’=G’’.

Figure 5. Shear elastic (G’) and viscous (G’’) moduli vs. temperature taken at constant corresponding frequencies of 0.1, 1 and 10 Hz for the sample PVC/40DINP

Certainly, such a transition cannot be properly defined considering only one frequency, as was highlighted by Winter (H.H. Winter Polym.Eng. Sci. 1987, 27, 1698). As frequency increases a shorter time is involved in the test, which should imply a higher temperature for the transition. Interestingly enough, our results of Figure 6, which shows the T (G’=G’’) transition of each sample at different frequencies, indicate that the effect of frequency is not indeed significant. The analysis of this unexpected result is out of the scope of this work.

Figure 6. Values of the crossover G’=G’’ temperature ,T (G’=G’’) , as a function of the applied frequencies for the different samples.

In the aim to focus rather on the effect of samples composition on the observed solid-liquid transition, we consider the results taken at a frequency of 1Hz, as can be seen in Figure 7. The corresponding T (G’=G’’) transition values are included in the table of the figure. Considering previous studies on the particulate flow of PVC formulations [41] we assume that the results are a consequence of the physical network formed by crystallites that only vanish at high temperatures, that is to say at temperatures at which G’’>G’ is observed. The crystallites network impedes reaching the terminal/flow viscoelastic zone. However, in the interval 180-210ºC (Figure 7), depending on composition, a cross-over of dynamic moduli occurs and the flow zone, characterized by G’’>G’, is attained. On the other hand, a G’ and G’’ maxima is observed upon increasing the temperature in the range 120-170ºC. As far as we know this result has not been reported in literature and can be explained considering the interaction between the plasticiser and PVC, which leads to structural changes during the fusion of the samples (G.C. Portingell in Particulate Nature of PVC: Formation, Structure and Processing Edited by G. Butters, 1982 Applied Science Publishers, London)

The reviewer remarks the unexpected G’ and G’’ maxima in temperature scans and asks for an explanation.

Answer: For sure these maxima are surprising and, to our knowledge, have not been reported before. In the revised version we offer an explanation based on the particular interactions between plasticisers and PVC particles. In our opinion the following paragraph, included in the revised text, is sufficiently clarifying considering the scope of the paper:

On the other hand, a G’ and G’’ maxima is observed upon increasing the temperature in the range 120-170ºC. As far as we know this result has not been reported in literature and can be explained considering the interaction between the plasticiser and PVC, which leads to structural changes during the fusion of the samples (G.C. Portingell in Particulate Nature of PVC: Formation, Structure and Processing Edited by G. Butters, 1982 Applied Science Publishers, London)

The reviewer makes a general comment, based on her/his personal opinion, about the difficulties to directly compare rheological and printing information and highlights the relevance of thermal history.

Answer: We agree with this interesting reflection and this point of view is implicitly expressed along the text. More in particular we allude explicitly to the relevance of the thermal history in the following paragraph of the revised version:

As can be seen in Figure 4 materials were nicely printed at temperatures above 200ºC, but the printability of the samples was limited below this temperature. In fact, only the materials presenting high amounts of plasticizer can be printed at 180ºC. This is not initially expected, because the samples indeed flow in the capillary rheometer at this temperature, as can be seen in Figure 1. But we have to consider that the diameter of the nozzle is actually the half of that of the capillary rheometer. More relevant can be the different thermal history of the melt in both devices, the rheometer and the printing nozzle: in the former the measurements are made under settled isothermal conditions, whereas in the latter the data are probably taken under transient thermal conditions

The reviewer considers our relaxation experiments confusing and rather naive, because they are carried out assuming a welding time of 1 s and we do not actually measure the filament temperature history from nozzle to welding bed.

Answer: Please consider the ambit and aims of the paper. We want to focus on the reliability of PVC formulations to bring about flexible 3D printing parts. The help of rheology for reaching this objective is, in our opinion, sufficiently clear considering the scope of the journal. We know that the rheology of 3D printing is making just its first steps and, certainly, our intention in this paper is not to make ex-cathedra pronouncements. We propose the Dahlquist criterion as a possible way to gain insight on the welding process, face to the criterion based on G´´>G’ which is demonstrated to be not valid. The choice of assuming a welding time of 1s is based on the few experimental and theoretical studies on the cooling process from nozzle to bed. In fact, we have carried a study, based on the paper of McIllroy and Olmsted (Polymer 2017), that demonstrates that 1s is adequate, but we do not consider suitable to include this research in this paper.

Reviewer 2 Report

I found the work interesting and worthy of publications. The authors wisely used the interplay between thermal, mechanical and viscoelastic characteristics of plasticized PVC to design a formulation that can be 3d printed using FFF technique. I would recommend this for publication but I do have one comment:

In the introduction, the authors need to justify why they used Plasticized PVC but not other alternatives such as TPU that has been proved to lead to fabrication of products with enhanced mechanical properties (high abrasion resistance and ductility). Also, different printable grades of TPU are currently being used. 

And some minor points:

Line 11: Define FFF or use Fused Filament Fabrication instead of the abbreviation.

Line 34: Remove the extra comma at the end of line after "applications,"

Line 96: Remove the extra point after (DINP)

Author Response

Dear Editor

We are pleased in submitting to your consideration for publication in MATERIALS a revised version of our manuscript entitled Searching for rheological conditions for FFF 3D printing with PVC based flexible compounds.

All the concerns and requirements of the reviewers have been contemplated and the manuscript has been modified accordingly (changes are highlighted in red in the text). Detailed corrections are listed as follows (the sentences given in Italic corresponds to the changes introduced in the revised manuscript):

Reviewer 2:

The reviewer considers necessary to justify why other alternatives for ductile polymers, such as TPU, are not included. He/she also remarks some grammatical errors in the text.

Answer: The errors have been corrected. The relevance of TPU in 3D printing is highlighted in the Introduction in the following corrected sentence:

Polycarbonate (PC), poly (ether imide) (PEI), Nylon 12, thermoplastic polyurethane (TPU) and other primarily amorphous thermoplastic polymers are also used to achieve extremely complex geometries. Except TPU, which allows 3D printing of ductile parts, currently the polymers used are brittle, which limits their applications for certain purposes. The exponential technological progress and the growing demands of the market yield the higher motivation to develop new materials for FFF. Among these new materials, the inclusion of more polymers (in addition to TPU) flexible and ductile is being considered day by day.

Reviewer 3 Report

This work is acceptable. 

Author Response

There are not comments or suggestions for authors.

Round 2

Reviewer 1 Report

The authors satisfactorily answered the criticisms.

Only one point remains to me: the shear rates (at least estimates!) in the nozzle are still not quantified. Please provide numerical values

Author Response

All the concerns and requirements of the reviewers have been contemplated and the manuscript has been modified accordingly (changes are highlighted in red in the text). Detailed corrections are listed as follows (the sentences given in Italic corresponds to the changes introduced in the revised manuscript):

Reviewer 1

The reviewer states that the shear rates in the nozzle are still not quantified and he/she encourages to provide numerical valuesAnswer: This has been clarified in the revised version including the following paragraph:

The corresponding shear rate limits are 2000 s-1 and 2 s-1 for the 0.4 mm nozzle and 666 s-1 and 0.66 s 1 for 1.2 mm nozzle.
